# Synthesis of Sorption Materials from Low Grade Titanium Raw Materials

**DOI:** 10.3390/ma15051922

**Published:** 2022-03-04

**Authors:** Lidia G. Gerasimova, Marina V. Maslova, Ekaterina S. Shchukina

**Affiliations:** Tananaev Institute of Chemistry of Kola Science Centre, Russian Academy of Sciences, 26a Fersman Street, 184209 Apatity, Russia; l.gerasimova@ksc.ru (L.G.G.); m.maslova@ksc.ru (M.V.M.)

**Keywords:** low-titanium raw materials, acid treatment, titanium-containing precursors, titanium sulfate, hybrid compositions, sorbents

## Abstract

In this paper, a universal technology is proposed for processing low-titanium mineral raw material—apatite-nepheline ore waste, including its treatment with sulfuric or hydrochloric acid in a two-stage mode with a sequential increase in the concentration. This technique allowed us to remove nepheline and apatite in the first stage and achieve a titanium mineral content of TiO_2_ of more than 30%; in the second stage, we were able to convert the titanium into its precursors—titanyl sulfate monohydrate TiOSO_4_·H_2_O and a hybrid rutile-silica composition. The key stage in the sorbent synthesis is the reaction of the precursor with a phosphoric acid solution. The preferred sequence of operations begins with the mechanical activation of the precursor, causing morphological changes in it, and subsequent treatment with phosphoric acid at different concentrations under atmospheric conditions and in an autoclave, accompanied by phase transformations. Conditions for the chemical reactions which regulated the composition and structure of the final product and, accordingly, its sorption activity were found. With the help of XFA, the phase compositions of the sorbents were identified, including the individual crystalline phase α-TiP obtained from TS or the crystalline phase αTi(HPO_4_)_2_∙H_2_O, which is in an amorphous silica matrix obtained from a rutile–silica composition.

## 1. Introduction

Russia ranks second in the world in terms of its proven reserves of titanium raw materials [1,2,3,4]. At the same time, not a single titanium deposit is being developed in the country, and the demand for raw materials and processing products is mainly satisfied through imports. Moreover, hundreds of thousands of tons of titanium are annually written off from the State Balance Sheet. Two-thirds of its raw material reserves are unconventional geological-industrial and mineral deposits and ores (titanomagnetite, sphene, leucoxene, perovskite). The Kola Peninsula, which is part of the Arctic Zone, has huge reserves of minerals, including titanium and titanium–rare metal ore deposits. Among these are the operational Khibiny deposit of apatite-nepheline ores, the Lovozersky loparite deposit, and the Afrikand deposit of perovskite ores. These deposits have real prospects for long-term industrial operations, since they are among the richest in terms of titanium and rare metals resources in Russia [5,6].

The Khibiny deposits of complex apatite-nepheline ores (ANO) are a unique natural phenomenon. Five main rock-forming minerals are regulated in the ore: apatite, nepheline, sphene, aegirine, and titanomagnetite (Table 1). Therefore, during mining and processing, there are complex problems associated with the need for a more complete extraction of the targeted concentrates in the context of deteriorating ore quality and the complexity of the integrated processing technologies.

Sphene and titanomagnetite are the rock-forming minerals of the Khibiny apatite-nepheline ores. Their contents in the ore are 2–2.5% and 1–1.5%, respectively. Titanomagnetite is extracted from apatite flotation tailings by magnetic separation, and sphene is extracted from nepheline flotation tailings by a rather long process, which provides a yield of the mineral of only 20–30% in the concentrate. A key aspect of most schemes used for the enrichment and separation of concentrates is the choice of flocculant [7,8]. The highest extraction of sphene in the concentrate from the foam product of the nepheline plant through the flotation method was achieved by the Mechanobr Institute technology (extraction of 55.2% at a content of titanium dioxide of 26%). The calculated content of titanium dioxide in the sphene mineral is ~37%. The expensive and scarce cupferon was used as a collecting reagent and the flotation concentrate was further processed by means of wet electromagnetic separation. Schemes for sphene concentrate production using high-frequency settling or screw separators were not considered promising due to the fact that the minerals contained in the apatite flotation tailings have almost the same size and insignificant differences in density, except for titanomagnetite. Wet magnetic separation in a weak magnetic field is currently used to produce sphene concentrate in order to remove the highly magnetic component of nepheline flotation foam (titanomagnetite) slurry. Classification in a screen reduces the total apatite content of pulp for chemical treatment and removes the coarse fraction. Strong-field wet magnetic separation is used to separate out the bulk of weakly magnetic minerals. Chemical cleaning is used to remove nepheline and apatite with dilute sulfuric acid. Dry magnetic separation finally separates the magnetic and non-magnetic fractions to produce chemically purified sphene and aegirine concentrates.

At present, apatite and a small amount of nepheline are produced during ANO enrichment. At the same time, little attention is paid to the possibility of the associated extraction of mineral concentrates that are not currently targeted. The intensive development of ANO mineral resources is accompanied by the formation of significant amounts of human-made enrichment waste, which are sent to dumps in the form of fine powders, thereby polluting the environment. Meanwhile, it is of practical interest to extract from this waste the mineral sphene–calcium titanosilicate (CaTiSiO_5_), which is 40% TiO_2_. The sphene concentrate obtained during the complex enrichment of apatite-nepheline ores does not contain iron, which significantly complicates the process of processing iron-containing ores, and, despite its low titanium content, could be an alternative to traditional raw materials. However, modern schemes for the separation of sphene concentrate from current production waste are ineffective and do not provide a high degree of mineral recovery [9]. In addition, enterprise interest is optimized for the demanded mineral concentrates. Therefore, the creation of a large-scale production method for sphene concentrate remains problematic.

Meanwhile, in many publications [10,11,12,13] it has been shown that the use of sphene concentrate is of scientific and practical interest for obtaining materials that contain little functional titanium, such as non-toxic tanning agents, as well as nanodispersed titanium dioxide, which can be used for various purposes, including in thermoregulated insulating materials, synthetic sorbents, pearlescent pigments, and photocatalytic catalysts. These materials are used in industrial areas for the manufacturing of products for civil and defense purposes. Their absence on the domestic market makes the successful development of advanced industries impossible. Within the framework of the small-scale production of the listed products, the payback period can be realized in a short time due to their high cost.

Thus, titanium-containing sorbents are included in the list of important inorganic functional products that are used in various regions, including in the Arctic. Recently, interest in the use of titanium phosphates as ion-exchange materials capable of effectively purifying aqueous solutions from the radionuclides and cations of non-ferrous heavy metals has increased. The structure of titanium phosphates and, accordingly, their functional properties—in particular, their sorption capacity—depend on the conditions used for the synthesis of the phosphate precursor: the initial ratio of the reagents, the state of ions in the initial solutions of these reagents, the methods and conditions of their precipitation, the duration of the gel maturation, and the conditions of washing and drying [14,15]. The temperature and acidity of the medium and the synthesis time are the main factors that determine the structure and properties of titanium phosphates [16,17,18]. Most often, the synthesis of the titanium phosphate precursor is carried out by precipitation from solutions of titanium (IV) salts, into which orthophosphoric acid is added [19,20]. Another method is based on the treatment of as-precipitated titanium (IV) hydroxide with orthophosphoric acid [21]. Less commonly, synthesis is carried out based on solutions with a low concentration of reagents (the sol–gel method) [22].

Titanosilicate sorbents (TSS) of the ion-exchange type that have a mineral-like framework structure are of interest; these are superior to the aluminosilicates (zeolites) widely used for sorption in terms of their thermal and radiation stability, selectivity, and higher sorption characteristics [23,24], particularly relative to single- and double-charged radionuclides. There are known industrial brands of TSS—for example, ETS-4 [25,26] and ETS-10 [27], which have structures similar to that of the mineral zorite, Na_8_Ti_5_[Si_12_O_14_](O,OH)_·_11H_2_O, and IONSIV IE-911, which has a similar structure to that of the mineral sitinakite, Na_2_KTi_4_Si_2_O_13_(OH)·4H_2_O [28]. In laboratory conditions, synthetic analogs of the mineral ivanyukite were obtained: Na_4_(TiO)_4_(SiO_4_)_3_·4–6H_2_O [29]. The specificity of their structure and the presence of cavities in the framework provide a high rate of ion-exchange processes in sorbent-sorbate systems.

The search for new raw materials, including waste of a technogenic nature, and the creation of innovative waste-free technologies based on them (which can surpass the capabilities of known methods in terms of their technological, environmental, and economic significance) are the tasks of this paper. At the same time, a positive result is achieved when there is an interconnection in the chain of raw materials → technology → product structure → properties.

The purpose of this research is to study the conditions of the acid treatment of unconventional titanium-containing raw materials to obtain precursors for the synthesis of effective sorbents.

## 2. Materials and Methods

### 2.1. Materials. Characteristics of the Raw Material Source

A polymineral material (sphenite ore-SO), derived from apatite-nepheline rocks from the Khibiny alkaline massif located on the Kola Peninsula, was chosen as the raw material source for obtaining the titanium-containing precursors in this research [30,31].

Sometimes there are areas called sphenite socket-lenses in which the concentration of sphene reaches 80% [32]. Figure 1 shows photographs of ore samples taken from the Koashva deposit [30]. The size of the sphenite ore blocks in some places reaches dozens of meters, which makes it possible to organize their selective extraction during the development of apatite deposits.

Such blocks of sphene can be easily separated from the total ore mass. In this case, the extraction of the sphene can reach 60–70% of the original ore. The mineral sphene contains, in wt%, TiO_2_ 39 ± 1, SiO_2_ 30.3 ± 0.7, CaO 27 ± 1, Al_2_O_3_ 0.2 ± 0.2, Fe_2_O_3_ 0.9 ± 0.3, Na_2_O 0.6 ± 0.3, Nb_2_O_5_ 0.7 ± 0.1, TR_2_O_3_ 0.5 ± 0.2 [31]. The low content of chromophoric impurities in the sphene and, in particular, iron greatly simplifies the acidic schemes for extracting titanium (IV) from the block [33]. Additives of niobium and rare earth elements, which pass into the titanium-containing precursors along with titanium, as a rule serve as modifiers of the finished product by adjusting its structure or properties—for example, its thermal stability [34] and photocatalytic activity [35].

For this research, we used SO, which is a mixture of minerals. Sample mineralogical analysis in wt%: titanite—49.4; apatite—14.3; nepheline—17.6, titanomagnetite—2.5. The rest were dark-colored minerals (aegirine and feldspars). The crushing of the SO was carried out in a ball mill for 10 h. A metal sieve was used to separate the fraction of the material that was less than 63 μm. In the fine fraction, the content of the three main minerals was, in wt%, 59.9, 15.4, and 17.0. This product was used in the further research.

### 2.2. The Acid Processing of Sphene Concentrate

Schematic diagrams of the SO fine fraction acid treatment with the formation of titanium-containing precursors in various compositions are shown in Figure 2. The removal of the minerals (apatite and nepheline) was carried out by treating the SO with diluted sulfuric and hydrochloric acids (chemical enrichment). The acid used was determined by taking into account the subsequent technological operations that would be carried out—in particular, the operation used for the decomposition of the titanite concentrate obtained during the chemical enrichment. Thus, in the first stage of processing, titanite concentrate (TC) was obtained; in the second stage, titanium-containing precursors were obtained in the form of titanium sulfate salt—e.g., titanyl sulfate monohydrate, TS, or hydrated titanium-silicon precipitate, TSP. The reagents used were H_2_SO_4_ 93.0%, HCl 37.5%, and chemically pure Na_2_CO_3_ (Reachem, Moscow, Russia).

To carry out the first stage, a sample of SO weighing 400 g was taken, placed in a glass beaker, and treated with a solution of sulfuric (50–100 g∙L^−1^ H_2_SO_4_) or hydrochloric (50–75 g∙L^−1^HC) acid at the ratio of M_sa_:V_a_ = 1:4 (where M_sa_ is the mass of the SA and V_a_ is the volume of the acid) at a temperature of 20 °C–50 °C for 2 h. The treated precipitate was filtered using a Buchner funnel, then washed with purified water from the mother liquid.

The second stage of acid treatment (decomposition of the SA obtained during the purification of the SO) was carried out in a glass three-necked flask with a volume of 500 cm^3^, equipped with a thermometer and a reflux condenser. The concentrate was preliminarily ground in a porcelain ball mill at a mass ratio of concentrate:balls = 1:7 for 6 h. Afterwards, the milled material was divided into three fractions, <28 μm, 28–63 μm, and 63–100 μm, using sieves. Sulfuric acid with a concentration of 450–600 g∙L^−1^ H_2_SO_4_ was poured into the flask, and a sample of SC was loaded in an amount corresponding to the ratio of M_sa_:V_a_ = 1:3.5. The reaction mass was kept at a temperature of 108 °C–112 °C for a period of 10 h. After that, it was filtered under a vacuum with the separation of the Ca-Si residue. Concentrated sulfuric acid was added to the filtrate to achieve a H_2_SO_4_ content of 850–900 g∙L^−1^, then heated to boiling (140 °C–145 °C) and kept in that state for 5–6 h. Under these conditions, titanyl sulfate monohydrate, TiOSO_4_ H_2_O (TS), will crystallize [36]. The chemical composition of TS was found to be as follows, in wt%: TiO_2_–44.91, SO_4_^2^^−^–53.96.

When using hydrochloric acid for the decomposition of the SA, its concentration was changed from 25 to 35% HCl, the temperature was in the range of 101–105 °C, and the ratio of M_sa_:V_a_ = 1:3.5. The holding time of the reaction mixture under the above conditions was 10 h. The acid suspension obtained during the decomposition was cooled, then the titanosilicate precipitate (TSP) was filtered off under a vacuum before being washed with cold water and dried in air (20 °C–25 °C). The hydrochloric acid filtrate was utilized to obtain crystalline calcium chloride and recycled hydrochloric acid (Figure 2).

### 2.3. Methods of Obtaining Titanium-Containing Products

To obtain a titanium phosphate sorbent (TiP), 20 g of TS was loaded into 50% H_3_PO_4_ solution under constant stirring at 60 °C, and the resulting suspension was kept for 3 h. The TiO_2_:P_2_O_5_ molar ratio was 1:1. The solid obtained was separated by filtration and washed with H_2_O at a solid to liquid ratio of 1:20.

For the synthesis of the silicon-containing titanium phosphate–TPS, TSP powder was used without preliminary grinding at Ssp 24.3 m^2^/g or after fine grinding (mechanoactivation, MA) in a planetary mill of the Pulverisette-7 type (Fritsch, Idar-Oberstein, Germany), Ssp 39.2 m^2^/g. The chemical composition of the milled sample was determined using a FT IR 200 spectrophotometer (Perkin Elmer, Waltham, MA, USA). The specific surface area of the samples was determined using a MICROMERITICS FLOWSORB II 2300 (Micromeritics GmbH, Norcross, GA, USA). The weighed portion of the powder was loaded into orthophosphoric acid (H_3_PO_4_ 87% Grade H Reachem, Moscow, Russia) at 50% H_3_PO_4_. The consumption of the components corresponded to the mass ratio of TiO_2_:P_2_O_5_ = 1:2. The mixture was kept at a temperature of 20 °C and 50 °C with stirring for 2 h. These conditions led to the leaching of the titanium and silicon to the liquid phase (hydrochloric acid), followed by the formation of a TPS precipitate. The calcium remained in the liquid phase due to its high solubility under these conditions. The precipitate was washed with water to remove the “acid” mother liquor and dried for 24 h at 65 °C–70 °C [37].

The method of TSP mechanical activation (MA) was also used to obtain the titanosilicate sorbents (TSS). The procedure used for the experiments was as follows. Titanium balls with amounts of 30 g and 3 g of TSP powder were loaded into a planetary mill jar (40 mL). The powder was crushed while rotating the drum at a speed of 700 rpm for 0.5 h; then, 3 g of sodium hydroxide (NaOH pure grade Reachem, Moscow, Russia) and 1 g of distilled water were added. The drum rotation and grinding continued for 1 h. A total of 35 mL of H_2_O was added to the crushed mixture, which was placed in an autoclave. The second experiment was the same, only differing in that NaOH was mixed with TSP before grinding. Hydrothermal synthesis was carried out at a temperature of 200 °C for 48 h. The resulting precipitate was separated by filtration, then washed with water and dried at 75 °C.

### 2.4. Characterization

The phase composition of the solid phases was determined with the XRD-6000 (Shimadzu GmbH, Tokyo, Japan) diffractometer with a Cu_Kα radiation source. Particle morphology was determined with the Philips XL 30 (F.E.I. Company, Hillsboro, OR, USA) scanning electron microscope. The surface properties of the produced samples were determined with the TriStar 3020 (Micromeritics GmbH, Norcross, GA, USA) device using the BEI and BJH methods based on nitrogen adsorption/desorption. The composition of the solid phases was studied by way of X-ray fluorescence analysis with the MAKS-GV spectroscope (Spectroscan, Saint Petersburg, Russia). TG-DSC data were received on an STA 409 PC/PG (NETZSCH, Germany) thermal analysis instrument at a heating rate of 10 °C·min^−1^ under high-purity N_2_ flow (20 mL·min^−1^). The Fourier transform infrared spectra (FT-IR) of the samples were obtained on a Nicolet 6700 (Thermo Fisher Scientific, Waltham, MA, USA) using the KBr method.

For the sorption study TP, TSS, and TPS were selected. The main components of the liquid radioactive waste of nuclear power plants are the long-lived radioisotopes of Cs^+^, Sr^2+^, and Co^2+^. To provide the scientific background of LRW purification on the materials obtained, the sorption capacity of the solids with respect to the chosen cations was established under static conditions. The sorption capacity was established under static conditions at a solid to liquid phase ratio of 1:200 and with a duration of contact between the sorbent and sorbate of 24 h. The cations in the model solutions had a concentration of 1 g/L Me. A calculation was carried out according to the formula E = (Cн−Cp)∙V∙m^−1^ mg/g, where Cн and Cp are the initial and equilibrium concentration of the cations, mg/g is the model solution, and V is the volume of the solution; the units used were ml, m (mass), and g. The content of the cations in the solutions before and after sorption was determined using an ELAN 9000 DRC mass spectrometer (PerkinElmer Life and Analytical Sciences, Shelton, CT, USA).

## 3. Results

Taking into account the available data on the solubility of apatite [38] and nepheline [39] in acidic media, the conditions for the chemical enrichment of the studied SO were chosen. Below are the reactions that comprise the process mechanism:

Apatite:(1)$$Ca_{5}{\left(PO_{4}\right)}_{2}F+5H_{2}O\to 5CaSO_{4}↓+3H_{3}PO_{4}+HF$$
(2)$$Ca_{5}{\left(PO_{4}\right)}_{2}F+10HCl\to 5CaCl_{2}+3H_{3}PO_{4}+HF$$

Nepheline:(3)$$\left[(KNa{)}_{3}\right]{\left[AlSiO_{4}\right]}_{4}+8H_{2}SO_{4}\to 2{\left(NaK\right)}_{2}SO_{4}+2Al_{2}{\left(SO_{4}\right)}_{3}+4H_{2}SiO_{3}+4H_{2}O$$
(4)$$\left[(KNa{)}_{3}\right]{\left[AlSiO_{4}\right]}_{4}+16HCl\to 3NaCl+KCl+4AlCl_{3}+4H_{2}SiO_{3}+4H_{2}O$$

The efficiency of removing apatite and nepheline impurities from SO (Table 2) increases with the use of hydrochloric acid. The interaction of apatite with sulfuric acid is accompanied by the formation of crystalline calcium sulfate (Reaction 1), which covers the SO grains and prevents (inhibits) the dissolution of apatite and nepheline minerals. This is evidenced by the data obtained using a scanning microscope (Figure 3a). When treated with a solution of hydrochloric acid, calcium from the apatite goes into the solution according to the reaction (2); therefore, the surface of the crystals has clear edges and there are practically no shells from the neoplasm on the surface (Figure 3b). An increase in the acid concentration and process temperature leads to the opening of fine particles of titanite, which is evidenced by the presence of titanium (IV) in the acidic liquid phase at up to 3–4 g·L^−1^ in TiO_2_.

The most favorable conditions for the first stage should be considered: for the sulfuric acid method, C_H2SO4_ at 70–75 g·L^−1^; for the hydrochloric acid method, C_HCl_ at 50–55 g·L^−1^. In these conditions, the temperature will not be more than 40 °C. Compliance with these conditions will allow one to obtain a concentrate (SC) containing 80–85% titanite. The SC composition is shown in Table 3.

The process of sphene interaction with sulfuric acid proceeds according to the reaction:(5)$$CaSiTiO_{5}+2H_{2}SO_{4}\to TiOSO_{4}+SiO_{2}\cdot xH_{2}O↓+CaSO_{4}↓+H_{2}O$$

When using hydrochloric acid for sphene decomposition, the following reactions take place:(6)$$CaSiTiO_{5}+4HCl\to TiOCl_{2}+SiO_{2}\cdot 2H_{2}O↓+CaCl_{2}$$
(7)$$TiOCl_{2}+H_{2}O\to TiO_{2}\cdot H_{2}O↓+2HCl$$

The mechanism of SC decomposition is the crystal lattice destruction of the mineral under the action of the reagent (acid) and an elevated temperature, with the transition of the components into the liquid phase. Due to their different levels of solubility in acidic media, solid reaction products are formed. In a sulfuric acid environment, a composite precipitate consisting of crystalline calcium sulfate and amorphous silica is formed. Titanium (IV) is concentrated in the liquid phase. The degree of titanium(IV) recovery into the liquid phase is significantly influenced by the dispersion composition of the SC (Figure 4).

The interaction of the milled concentrate with sulfuric acid proceeds through two successive stages. The first stage of the process (external diffusion) is a chemical reaction that occurs on the surface of mechanically activated mineral particles to form a shell consisting of solid reaction products. Titanium is leached into the sulfate liquid phase. The duration of this stage is approximately 60 min. As the amount of calcium silicate precipitate in the reaction mass increases, the process slows down, proceeding to a second stage (intra-diffusion) associated with reagent transport through this shell and the destruction of the mineral. The reaction rate at this stage is considerably slower than that at the first stage. After about 7 h, the reaction rate is reduced to a minimum and the system reaches a pseudo-equilibrium state.

The actual degree of the leaching of titanium(IV) into the liquid phase and, consequently, the degree of SC leaching with different particle dispersions varies from 60 to 90%. A graphical method was used to calculate the rate constant of the SC decomposition reaction [40]. Dependence graphs were drawn in coordinates—ln(C_TiO2_) = f(τ). In the analysis of these, the rate constants of the first and second stages of the process were determined (Table 4).

The concentration of sulfuric acid of 600 g·L^−1^ H_2_SO_4_ ensures the high stability of the titanium-containing liquid phase (titanium sulfate solution), in which Ti(IV) is in a molecularly dispersed state [41]. The composition of titanium sulfate solution in g·L^−1^ is TiO_2_ = 90–100 and H_2_SO_4_ = 480–520. This solution was used to obtain titanyl sulfate monohydrate at TiOSO_4_∙H_2_O (TS).

The powder X-ray pattern (Figure 5) indicates a prevalence of the titanyl sulfate phase, TiOSO_4_ H_2_O, in the sample. The chemical composition of the TS was established by a chemical analysis of the sample in wt%: TiO_2_, 38–39; SO_3_, 52.5–53.0; Nb_2_O_5_, 0.6–0.7; Fe_2_O, 0.2–0.25; CeO_2_, 0.12–0.15.

In the hydrochloric acid system TiO_2_-SiO_2_-CaO-HCl, calcium is highly soluble, while titanium and silicon are co-precipitated as hydrated oxides. The silicon, due to its lower solubility in comparison with titanium(IV), goes into the precipitate (silica) in the first place [42]. The titanium’s (IV) solubility in the hydrochloric acid medium depends on its concentration in the HCl system (Figure 6). When using 25% HCl during the boiling of the reaction mixture for 2.5 h, the degree of titanium (IV) transition to the hydrochloric acid liquid phase does not exceed 60%, after which time the titanium precipitate begins to form. With an increase in the concentration to 30 and 35% HCl, the degree of the titanium (IV) transition noticeably increases and reaches 93% and 96%, respectively. In these cases, the leaching rate (before the formation of the titanium precipitate) is higher in the case of 35% HCl. In all cases, the titanium (IV) is almost completely precipitated after a process duration of 10 h. The titanium–silicon precipitate (TSP) isolated during the decomposition of the SC was then washed with water to remove the hydrochloric mother liquid. Calcium was concentrated in the hydrochloric acid liquid phase, during which evaporation removes the free hydrochloric acid and crystallizes the precipitate in the form of calcium chloride.

Using the powder X-ray patterns, it was found that the TSP contains two phases: crystalline rutile and amorphous silica (see Figure 7a). Using a thermal analysis, the weight loss of the sample (Figure 7b) during its thermolysis was determined. The endothermic peak on the thermogram was conditioned on the removal of the physically adsorbed and coordinatively bound water. The weight loss in the temperature range from 52.2 to 129.7 °C was 5.21 wt%. Another broad exothermic peak at a temperature of 500–600 °C indicated the transformation of the titanium dioxide crystal lattice with the formation of an ordered rutile structure. The weight loss in the temperature range from 200 to 900 °C was 5.05 wt%. The total loss incurred during the TSP thermolysis was 10.26%. This was mainly associated with the loss of water.

Judging from the SE image of TSP particles on the microphotograph (Figure 7c), they are agglomerates of fragments with a fairly regular shape, which is characteristic of crystalline substances—in this case, rutile particles. Apparently, amorphous silica formed a shell coating on the surface of the crystals, which was confirmed by the infrared spectroscopy data (Figure 7d). It was not possible to determine the response of the titanium component of the spectra, probably due to the shielding layer of the shell.

Thus, the precipitate (TSP) isolated during the interaction of titanite with hydrochloric acid was a scarcely hydrated product of the composition (mol) TiO_2_∙1.8–1.85SiO_2_∙0.75–0.8H_2_O, containing the additions of niobium and iron.

The specific surface area of the TSP as well as the pore system indices were determined. The specific surface (Table 5) corresponds to 30.75 m^2^/g, and the main contribution to this value seems to be from the amorphous silica phase.

Based on the particle diameter index of the composition (D) determined by the adsorption and desorption of N_2_, we can state that on the surface of the composition particles, there are mesopores with a “bottle” configuration, as evidenced by the hysteresis loop in the relative pressure (p/p^o^) at 0.50–0.95 (Figure 8).

The obtained precursors were used for a synthesis of the titanium-containing sorbents.

### 3.1. Synthesis of the Titanium Phosphate Sorbent–TP

Since the bond between the sulfate groups and titanyl ion in TS was sufficiently strong, the substitution of SO_4_^2–^ groups by phosphate groups during the heterogeneous synthesis occurred slowly. The optimal time for the synthesis was determined by the experiments. It was found that an increase in the synthesis time led to a decrease in the phosphorus content in the final product (Table 6).

The XRD data indicate that that synthesis time of 3 h was sufficient for pure α-TiP formation, as confirmed by the presence of the TS phase in the XRD pattern. In 5 h, a pure alpha titanium phosphate phase was formed (Figure 9).

In contrast to the current methods of α-TiP synthesis, which require rigid synthesis conditions and a high reagent consumption, the new synthesis lasted 5 h at stoichiometric reagent consumption and no special equipment was needed. Earlier, we showed that the heterogeneous interaction of milled TS with 50% phosphoric acid yields titanium phosphates of Ti(HPO_4_)_2_·H_2_O compositions for 3 h [43]. In this investigation, we found that freshly made TS could be used for synthesis without milling. The synthesis time increased slightly from 3 to 5 h, but the energy-intensive grinding operation was excluded.

The textural properties of the material obtained are given in Table 7 and Figure 10.

According to the data obtained, both the surface area and total pore volume were small due to the crystalline structure of α-TiP. From the pore size distribution curve, the predominant pores in the obtained material were concluded to be narrow mesopores with a pore diameter ranging from 3 to 10 nm. The narrow pore size distribution curve allowed us to predict the selective properties of the sorbent. The experimental data showed that the complete removal of Cs^+^, Sr^2+^, and Co^2+^ ions from solutions was limited by the initial concentrations of ions. These were 5, 6, and 8 mmol∙L^–1^ for strontium, cobalt, and cesium, respectively. From the isotherms obtained, the static ion-exchange capacities at ambient temperatures were 230, 84, and 76 mgl∙g^–1^ for Cs^+^, Sr^2+^, and Co^2+^ ions, respectively.

### 3.2. Synthesis of the Composite Silicon-Containing Titanium Phosphate Sorbent–TPS

Silicon-containing titanium phosphate sorbent (TPS), like many other composite materials, can have improved physicochemical and operational characteristics [44]. The crystalline filler included in the TPS determines the functionality of the material, while the amorphous matrix ensures the homogeneity of the composition and its resistance to external influences [22].

Table 8 summarizes the experimental conditions of this study and the properties of the solid phases obtained.

The X-ray diffraction data demonstrate that the TSP does not react with phosphoric acid without mechanical activation. This is due to the low reactivity of the rutile modification of titanium dioxide in its composition. An increase in the reactivity of TSP after its pre-activation in a planetary mill was found only in the sample obtained at a temperature of 50 °C. After investigating its composition, two phases were identified: rutile and crystalline titanium hydrogen phosphate monohydrate, α-Ti(HPO_4_)_2_·H_2_O (Figure 11), which was formed according to the reaction scheme:(8)$$TiO_{2}+2H_{3}PO_{4}\to Ti{\left(HPO_{4}\right)}_{2}\cdot H_{2}O↓+H_{2}O$$

The bonding between the titanium(IV) and phosphate groups was rather strong, ensuring the formation of a crystal structure with a dense packing of titanium phosphate groups (Figure 12):

Morphological and phase transformations in various oxides during dry grinding even for a very short time (down to 3–5 min) were observed in many studies [45,46,47]. The mechanical activation of TSP particles leads to a decrease in particle size, which leads to the increase in the specific surface area by 30–40%. Mechanical activation also causes morphological and microdeformation changes in rutile crystal particles, as noted in [48]. As a result of the high-energy shock-sliding impact on solid particles of heavy balls, rutile acquires a surface charge, which enhances the interaction of the activated TSP powder with H_3_PO_4_. The presence of hydrogen phosphate groups in the obtained composite product provides its ability to absorb Cs^+^ and Sr^2+^ cations from solutions due to exchange processes (Table 8, samples 3 and 4). The extraction of cations Cs^+^ and Sr^2+^ in experiments with TPS samples in which there are no hydrogen phosphate groups apparently mainly occurs due to surface adsorption (Table 8, samples 1, 2) [49].

One of the factors determining the sorption activity of a material is its porosity. According to the results obtained, the adsorption-desorption isotherm has a hysteresis loop in the region of relative pressure of 0.4–0.6, which is typical for planar particles and is a combination of H1 and H3 types according to the IUPAC classification (Figure 13). The absence of a hysteresis loop in the low-relative-pressure region indicates the presence of wide micro- and narrow mesopores. At the same time, the mesopores are dominant. This fact is also confirmed by the pore distribution curves (Figure 14).

The shift in the maximum to higher values was due to an increase in the number of wide mesopores and the filling of micro and narrow mesopores. The combination of these results made it possible to present the synthesized samples as mesoporous products.

### 3.3. Synthesis of Titanosilicate Sorbent (TSS)

The X-ray imaging of the samples under study (Figure 15 and Figure 16) and the subsequent interpretation of the X-ray data allowed the kinetics of solid-phase transformation during mechanical activation to be traced and the influence of morphological changes on the chemical activity of the milled material in the subsequent synthesis of the titanosilicate product to be established.

If all the components of the mixture are loaded into the mill bowl at the same time, a gradual structural transformation of the solid phases will take place during the mechanical activation process. The particles are transformed on the surface by amorphization.

In the case of sodium hydroxide loading after one hour of TSP treatment, significant changes were observed in the solid-phase system, ranging from surface transformations of the mixture component particles to phase transformation with the formation of titanosilicate nuclei (Figure 15 and Figure 16). Thus, under the autoclave curing conditions of the crushed sample, the formation of the titanosilicate phase was accelerated and the final product crystallized almost completely as a new phase with a zorite structure, with the impure phase being rutile. The specificity of the transformations in the described experiment seemed to be due to the fact that, in the absence of sodium hydroxide, the mixture received significantly more energy per unit mass. By the time sodium hydroxide was added, the surface activity of the components was higher than that in the case of a single load.

The order in which the mixture components were fed to the milling process was reflected in the morphology of the surface layer of particles that underwent mechanoactivation and hydrothermal treatment in the autoclave. Figure 17 and Figure 18 show microphotographs of particles of the tested samples.

In the absence of sodium alkali after an hour of milling, the intensive amorphization of the particle surface was observed. Due to this, during the following mechanoactivation with NaOH, a layer of a porous new formation was formed on the surface—apparently, titanosilicate germs (Figure 17a). Under autoclave conditions, these contribute to the formation of the zorite crystal phase (Figure 18a).

In the system of TSP + NaOH, during mechanoactivation a delamination of the surface layer of particles was observed; this was intensified with the increasing duration of the process, apparently due to the formation of sodium silicate compounds (Figure 17b). At the same time, titanium particles were transformed to a much smaller degree, inhibiting the formation of zorite under autoclave conditions (Figure 18b) [50,51].

The main indicators of the surface and sorption properties of the TSS are given in Table 9 and Figure 19. Alkaline titanosilicate is characterized by a sufficiently developed particle surface (S_sp_–81 sm^2^/g). The total volume of the mesoscale pores was 0.18 sm^3^/g.

## 4. Discussion

In this research, a universal technology for the processing of low-titanium mineral raw materials was developed. These materials are a form of technogenic waste from apatite-nepheline ore enrichment, with the formation of products demanded by both industry and ecology—sorbents for the treatment of liquid radioactive waste and toxic effluents from non-ferrous metallurgy. By treating raw material with sulfuric or hydrochloric acid in two-stage mode with an increase in concentration, it was possible in the first stage to remove the acid-soluble minerals (nepheline, apatite, and concentrated titanium in the form of sphene concentrate) and, in the second stage, to convert the titanium from the mineral into titanium-containing precursors: titanyl sulfate TiOSO_4_·H_2_O (TS) and rutile-silica composition (TSP).

It was demonstrated that solid TS can be successfully applied as a precursor for titanium phosphate synthesis. The application of the TS makes it possible to markedly cut down the number of synthesis stages, the amount of effluents used, and the consumption of reactants compared to the known methods of TiP synthesis. It was shown that material obtained had a good sorption ability for Cs, Co, and Sr ions and could be used for liquid radioactive waste purification.

The preliminary mechanical activation of the TSP in the high-speed planetary mill Pulverisette-7 increased the S_sp_ index by 1.5 times (39.2 m^2^/g), which increased the speed and completeness of its interaction with the 50% H_3_PO_4_ at the mass ratio of TiO_2_:P_2_O_5_ = 1:2 and provided the formation of a crystalline compound with the formula α-Ti(HPO_4_)_2_∙H_2_O, which was distributed in the silica matrix.

The mechanically activated mixture of TSP and sodium alkali, in the process of the hydrothermal treatment at a temperature of 200 °C for 72 h, was converted into an alkaline titanosilicate with a structure similar to that of the mineral zorite, corresponding to the formula Na_8_Ti_5_[Si_12_O_14_](O,OH)_5_∙11H_2_O. The obtained materials showed rather high sorption properties in relation to the mono-divalent cations. The proposed technology significantly surpasses the known processes in terms of both its economic and environmental characteristics.

## 5. Conclusions

In this research, a sulfuric acid and hydrochloric acid scheme for the processing of low-titanium sphene concentrate was developed. This scheme makes it possible to purify sphene from associated minerals and concentrate the titanium contained therein. Using the sulfuric acid process, we converted the titanium present in a sphene concentrate to a salt, titanyl sulfate monohydrate TiOSO_4_·H_2_O. Then, using the hydrochloric acid process, we converted this to a precipitate consisting of rutile and a silicon composition. The possibility of using acids in recycling technology was thus shown. The possibility of using calcium silicate cake from the sulfuric acid processing of sphene concentrate as a filler in building materials was proposed. The hydrochloric acid filtrate from the hydrochloric acid process could be recycled to produce calcium chloride, which is used as a de-icing agent.

The possibility of obtaining sorption materials from the products of sphene concentrate processing was shown in this paper. α-TiP was obtained from titanium sulphate monohydrate, crystalline αTi(HPO_4_)_2_∙H_2_O was obtained from the rutile silicon composition after mechanical activation, and an alkaline titanosilicate with a structure similar to that of the mineral zorite, corresponding to the formula Na_8_Ti_5_[Si_12_O_14_](O,OH)_5_∙11H_2_O, was obtained. The sorption properties of the obtained materials were studied.

## Figures and Tables

**Figure 1 materials-15-01922-f001:**
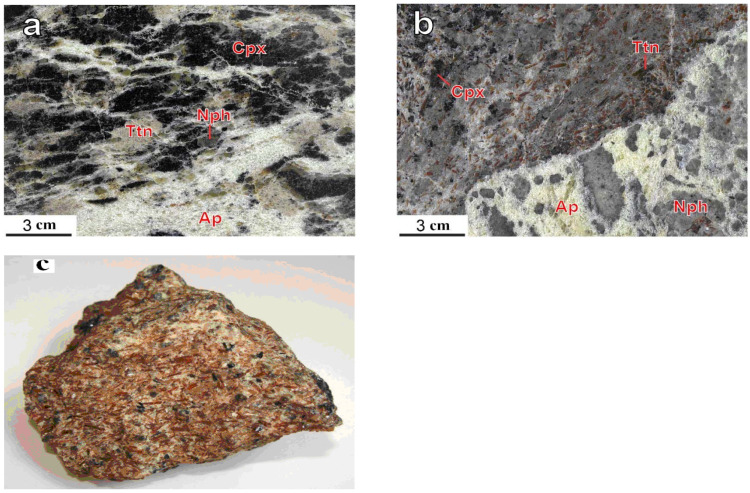
Titanite segregations in apatite-nepheline (**a**); apatite-titanite (**b**) ores of the Koashva apatite deposit, the Khibiny massif; (**c**)—“Sphenite” ore-SO. Ap—fluorapatite; Cpx—clinopyroxene; Nph—nepheline; Ttn—titanite.

**Figure 2 materials-15-01922-f002:**
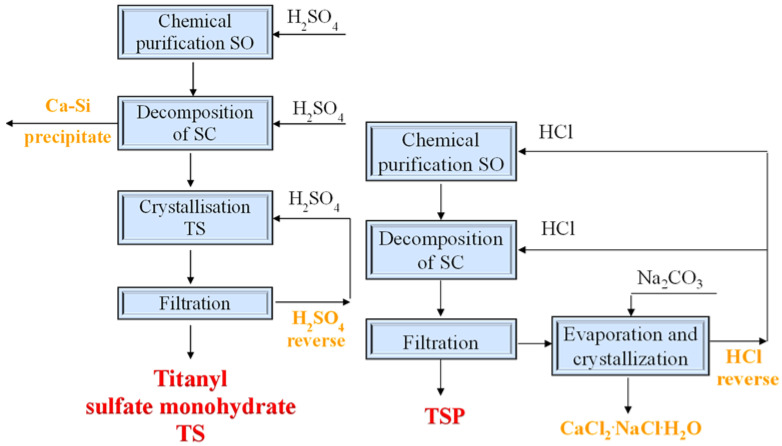
Technological schemes used for obtaining precursors for the synthesis of sorbents from titanium-containing SO: 1, sulfuric acid method; 2, hydrochloric acid method.

**Figure 3 materials-15-01922-f003:**
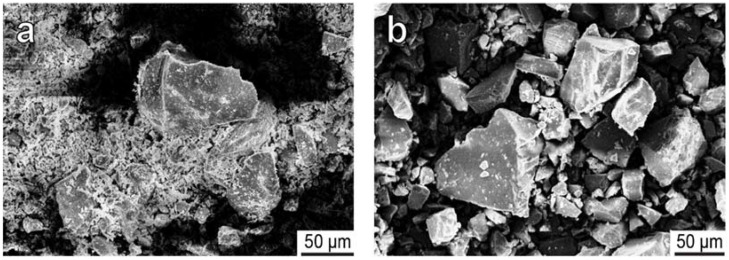
SE images of sphene grains cleaned by sulfuric acid (**a**) and hydrochloric acid (**b**).

**Figure 4 materials-15-01922-f004:**
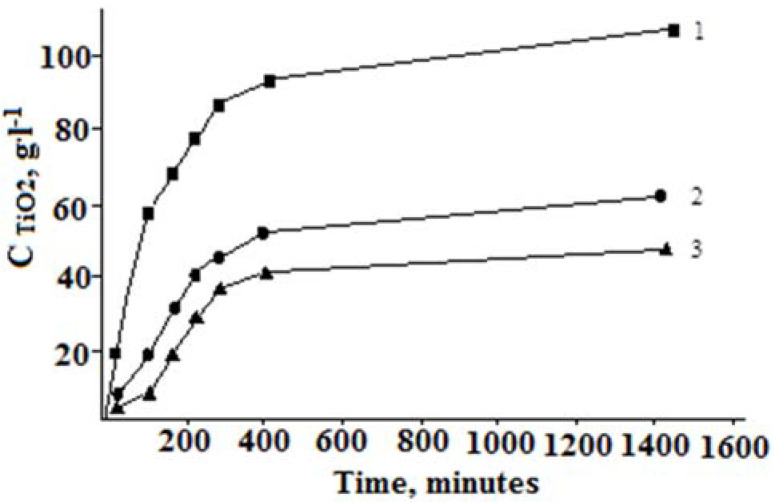
Influence of the dispersibility (μm) of SC particles on the liquid-phase titanium(IV) extraction. H_2_SO_4_ -600 g·L^−1^. Fraction of SC particles in microns: 1— < 28, 2—28–63, 3— > 63.

**Figure 5 materials-15-01922-f005:**
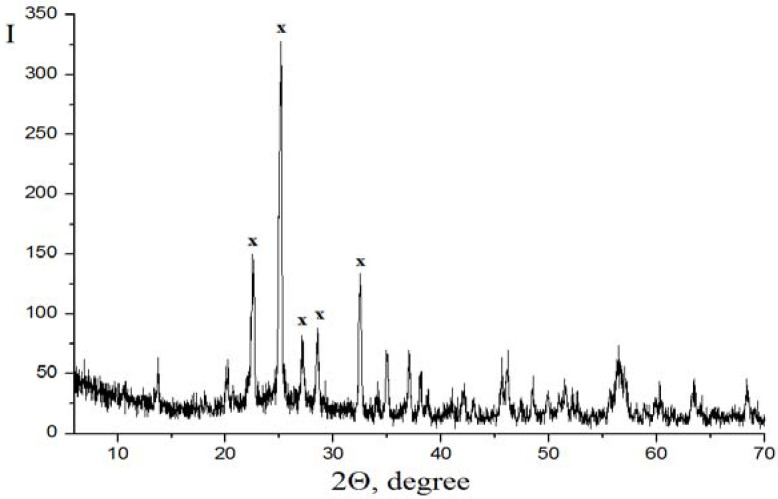
The powder X-ray pattern of titanyl sulfate monohydrate, TiOSO_4_·H_2_O (x), obtained from a sulfate solution after the leaching of SC.

**Figure 6 materials-15-01922-f006:**
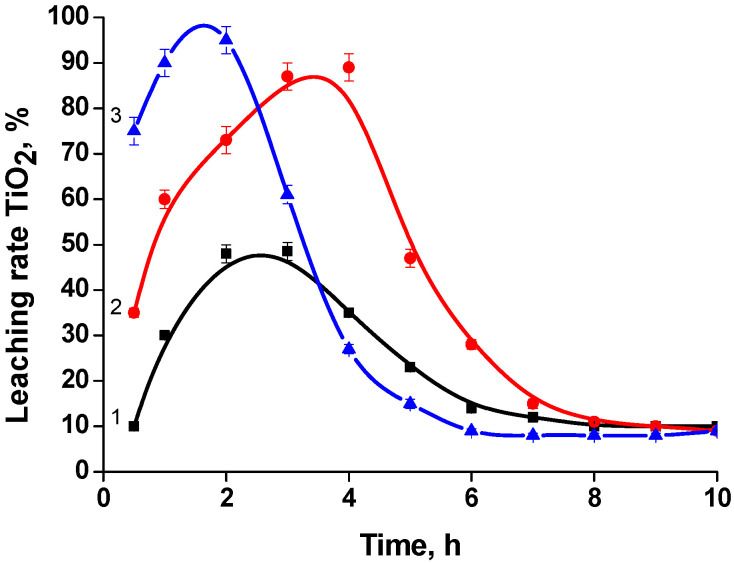
Effect of acid concentration (% HCl) on the degree of titanium (IV) transition to the liquid phase: 1–25; 2–30; 3–35; M_SC_:V_HCl_ = 1:3.5.

**Figure 7 materials-15-01922-f007:**
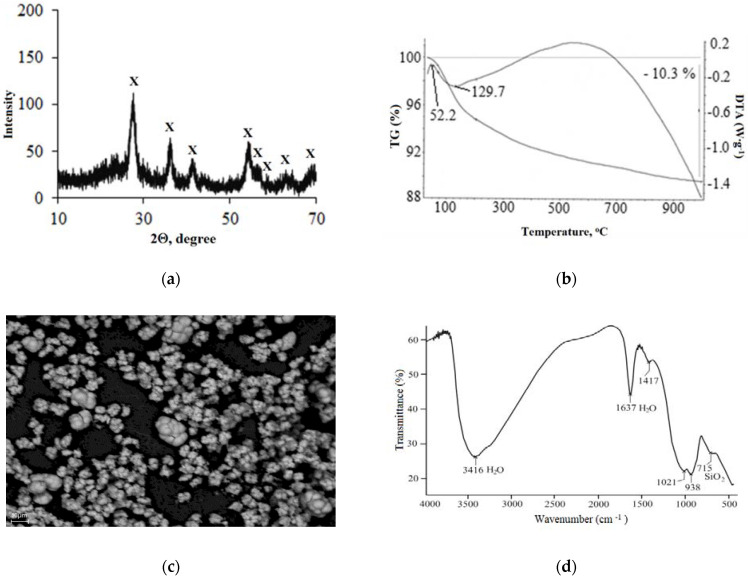
The powder X-ray pattern (**a**), thermogram (**b**), SE image (**c**), and FT-IR spectra (**d**) of the TSP sample (the sample was obtained by the leaching of SC with 32% hydrochloric acid). X: rutile.

**Figure 8 materials-15-01922-f008:**
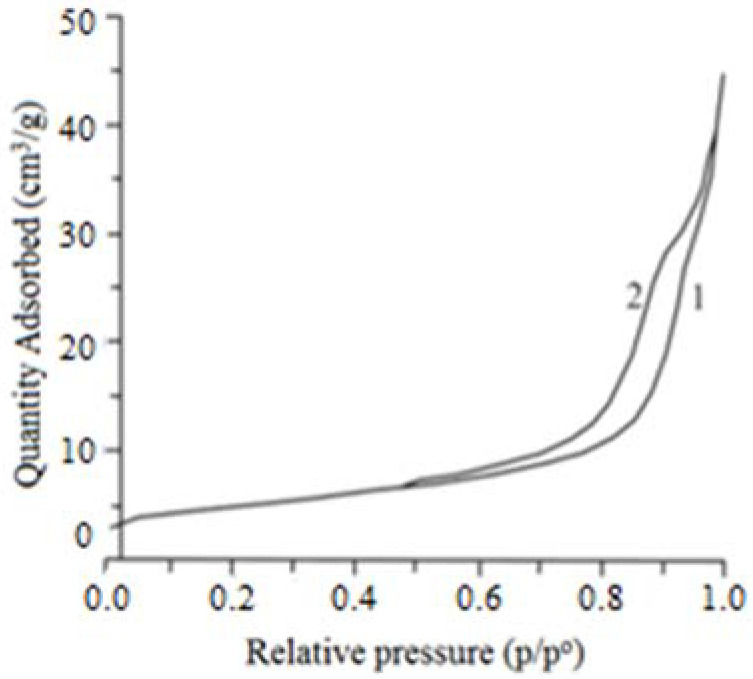
Characteristics of the pore system of TSP particles: 1, by N_2_ sorption; 2, by N_2_ desorption.

**Figure 9 materials-15-01922-f009:**
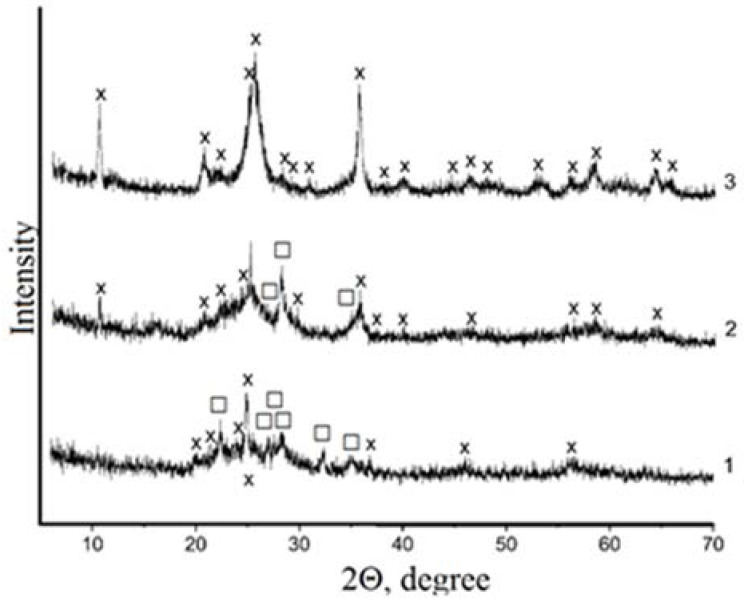
Powder XRD of TiP obtained by the decomposition of TS by 50% H_3_PO_4_. Synthesis time was 1 h (1), 3 h (2), and 5 h (3)**.** x-Ti(HPO_4_)_2_⋅H_2_O, □-TiOSO_4_⋅H_2_O.

**Figure 10 materials-15-01922-f010:**
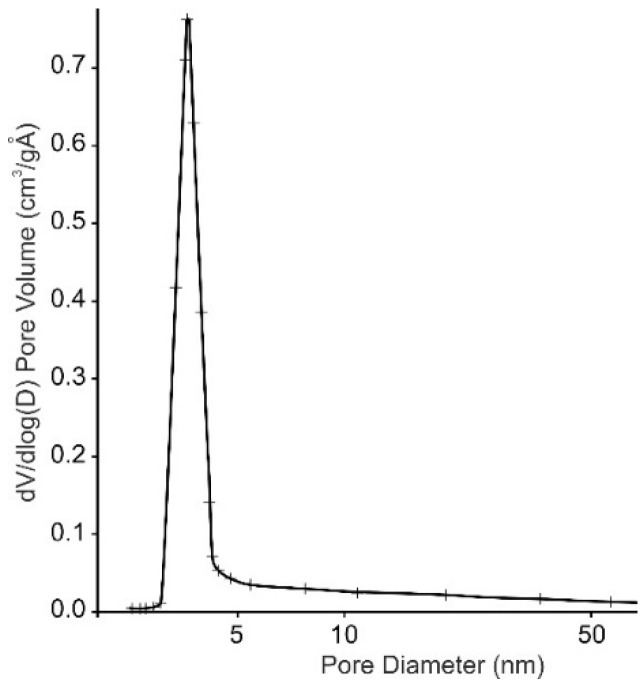
BJH pore size distribution plots derived from the desorption branch of the isotherm and α-TiP obtained.

**Figure 11 materials-15-01922-f011:**
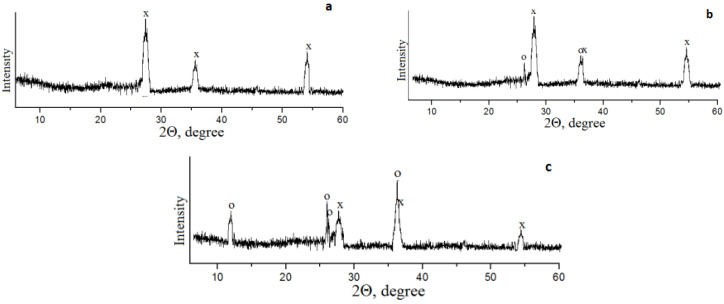
X-ray diffraction patterns of the TPS samples prepared by treating the TSP with 50% H_3_PO_4_: (**a**) without MA, (**b**) after MA at 20 °C, (**c**) after MA at 50 °C (atmospheric conditions). x-rutile; **○-**α-Ti(HPO_4_)_2_·H_2_O.

**Figure 12 materials-15-01922-f012:**
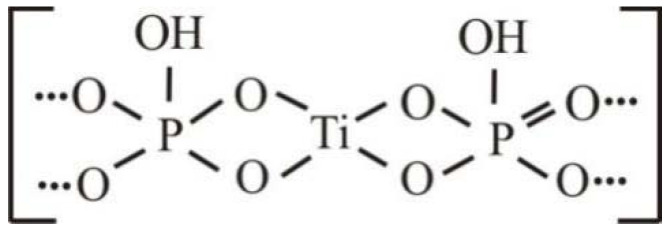
Titanium phosphate diagram.

**Figure 13 materials-15-01922-f013:**
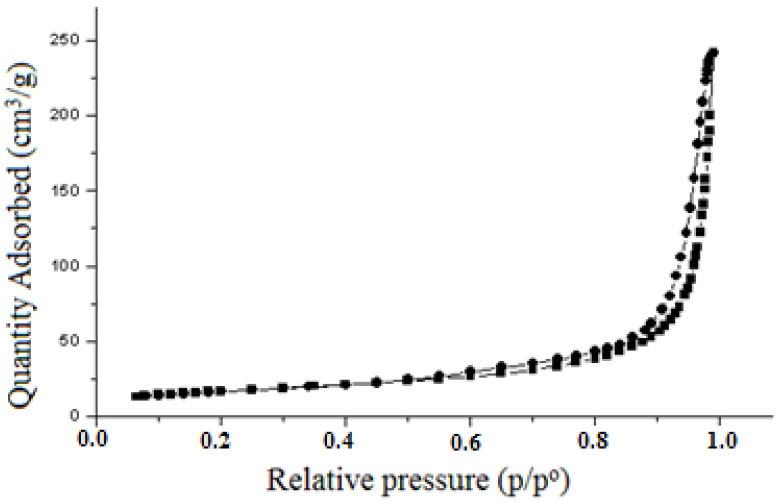
The adsorption–desorption isotherm for TPS obtained from sample 4 from Table 8.

**Figure 14 materials-15-01922-f014:**
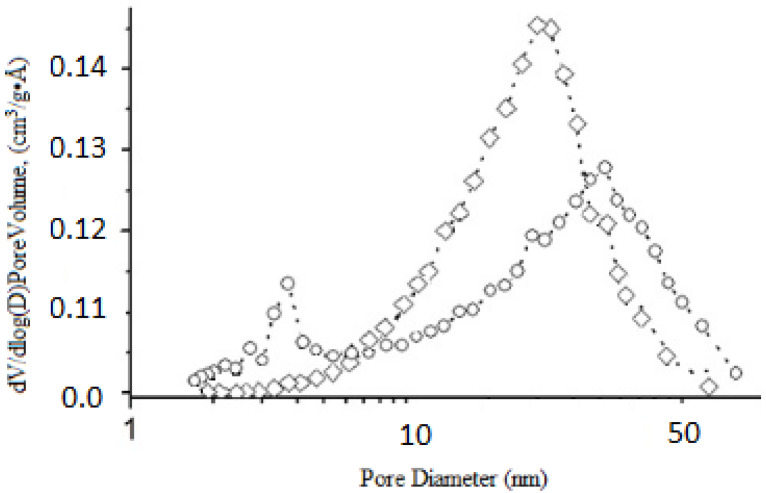
BJH pore size distribution plots derived from the desorption branch of the isotherm for obtained TPS samples from Table 8: □-2; ○-4.

**Figure 15 materials-15-01922-f015:**
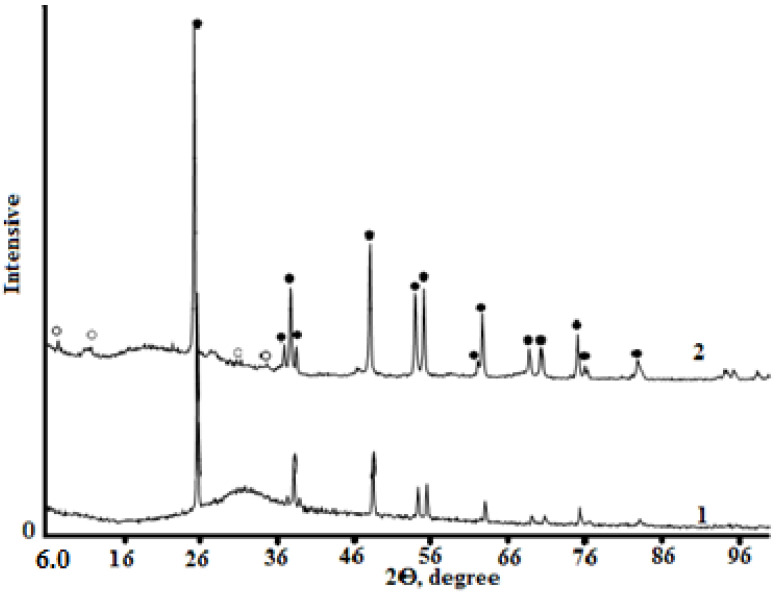
X-ray patterns of solid-phase samples after 1 h of mechanical activation (1) and subsequent autoclave treatment (2). The initial mixture initially contains NaOH. Phases: ○-zorite; ●-rutile.

**Figure 16 materials-15-01922-f016:**
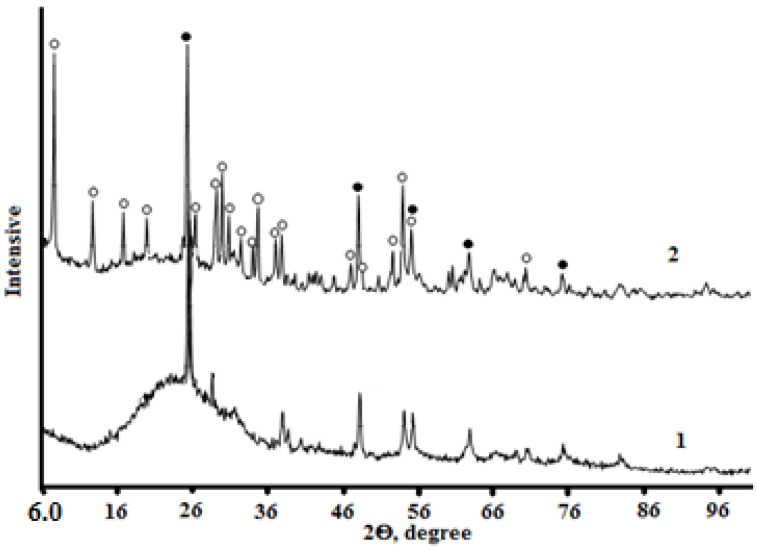
X-ray patterns of solid-phase samples after 1 h of mechanical activation (1) and subsequent autoclave treatment (2). The addition of NaOH to the mixture was carried out after 1 h of grinding. Phases: ○-zorite; ●-rutile.

**Figure 17 materials-15-01922-f017:**
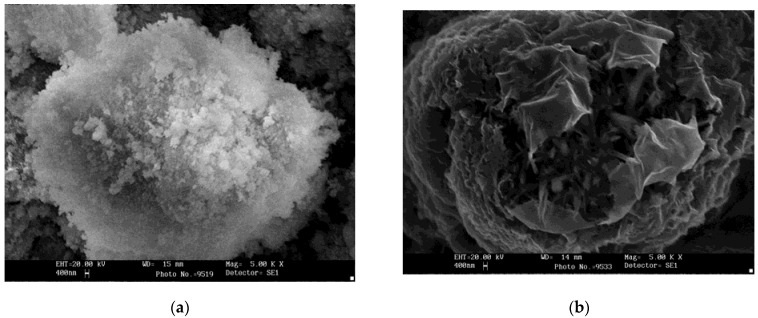
SE images of particles of mechanically activated mixtures of components: (**a**) sodium hydroxide added to the TSP after 1 h of mechanical activation; (**b**) mixture loaded simultaneously before mechanical activation for 1 h.

**Figure 18 materials-15-01922-f018:**
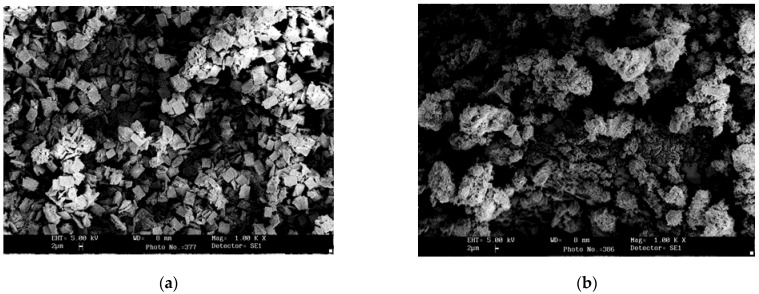
SE images of the product particles after 72 h of autoclave treatment. (**a**) Sodium hydroxide loading was carried out after 1 h of mechanical activation; (**b**) components loaded at one time.

**Figure 19 materials-15-01922-f019:**
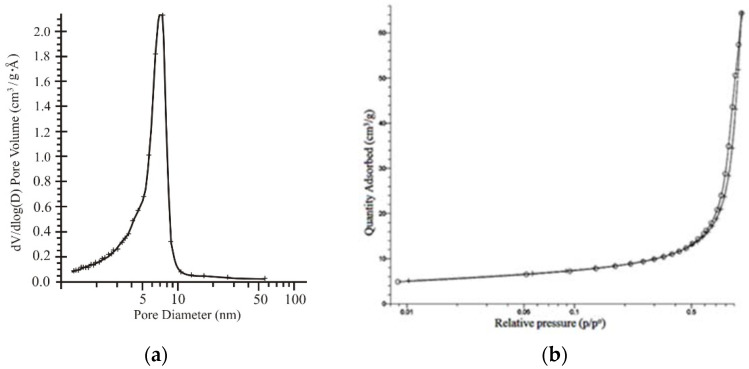
The adsorption–desorption isotherm for TSS obtained sample (a) and BJH pore size distribution plots derived from the desorption branch of the isotherm (b).

**Table 1 materials-15-01922-t001:** The main minerals composing ANO.

Mineral	Mineral Content in Ore, wt%	Mineral Formula
Apatite	33.7–35.0	Ca_5_(PO_4_)_3_F
Nepheline	40.6–42.2	(NaK)_2_OAI_2_O_3_·2SiO_2_
Aegirine	8.7–9.5	NaFeSiO_6_
Sphene	2.4–2.9	CaSiTiO_5_
Titano-magnetite	1.0–1.2	FeO·Fe_2_O_3_·TiO_2_

**Table 2 materials-15-01922-t002:** Conditions for the chemical enrichment of SO and the composition of the obtained SC.

Experimental Conditions		TiO_2_ at the Filtrate g∙L^−1^	Content of Components at Sphene Concentrate, wt%		
C_acid_, g∙L^−1^	T,°C		TiO_2_	P_2_O_5_	Al_2_O_3_
H_2_SO_4_					
50	20	traces	26.3	4.2	2.18
50	50	traces	29.9	2.8	2.01
75	20	0.25	30.4	3.0	1.23
75	50	0.98	32.6	1.6	1.12
100	20	1.55	32.8	3.1	1.25
100	50	3.93	30.4	3.6	1.33
HCl					
50	20	traces	29.8	3.2	1.71
50	50	0.76	33.2	1.4	1.50
75	20	0.96	31.9	1.9	1.11
75	50	2.78	30.4	1.8	1.10

**Table 3 materials-15-01922-t003:** Sphene concentrate composition.

**Components**	TiO_2_	P_2_O_5_	Al_2_O_3_	Fe_2_O_3_	Nb_2_O_5_	TR_2_O_3_
**Content, wt%**	31.8	1.38	1.21	0.35	0.42	0.30

**Table 4 materials-15-01922-t004:** Rate constants for the interaction of sphene with sulfuric acid (H_2_SO_4_-600 g·L^−1^).

Fraction of SC Particles, Microns	Rate Constant of Decomposition, min^−1^		K_1_/K_2_
	First Stage-1	Second Stage-2	
<28	1.8 × 10^−2^	5.522 × 10^−4^	32.5
28–63	1.5 × 10^−2^	3.795 × 10^−4^	39.5
63–100	1.2 × 10^−2^	2.291 × 10^−4^	52

**Table 5 materials-15-01922-t005:** Textural properties of the TSP.

Sample	S, m^2^·g^–1^	V_pores_, cm^3^·g^–1^ (Sorption)	V_pores_, cm^3^·g^–1^ (Desorption)	D, nm (Sorption)	D, nm (Desorption)
TSP	30.75	0.068	0.069	16.48	14.34

**Table 6 materials-15-01922-t006:** Composition of the solids obtained.

No Experience	Synthesis Time, h	Content, wt%		
		P_2_O_5_	TiO_2_	S
1	1.0	37.85	29.67	0.21
2	3	48.35	29.41	0.06
3	5	51.68	29.13	0.01

**Table 7 materials-15-01922-t007:** Textural properties of the α-TiP obtained.

Sample	S, m^2^·g^–1^	S_ex,_ m^2^_·_g^–1^	V_pores_, cm^3^·g^–1^	D_av_, nm
α-TiP	6.33	6.16	0.014	5.12

**Table 8 materials-15-01922-t008:** Effect of the synthesis conditions on the sorption properties of TPS.

No Experience	Experimental Conditions	S_sp_, m^2^·g^−1^	Static Capacity TPS, mg·g^−1^	
			Sr^2+^	Cs^+^
1	TSP without MA, T–20 °C	24.5	20.0	8.1
2	TSP after MA, T–20 °C	41.9	25.4	13.0
3	TSP without MA, T–50 °C	34.0	32.9	35.1
4	TSP after MA, T–50 °C	78.1	58.7 *	41.2 *

* For comparison of the static sorption capacity of zeolite SM-5, mg/g: Sr^2+^, 30; Cs^+^, 20.

**Table 9 materials-15-01922-t009:** Textural and sorption properties of the obtained zorite.

Surface Properties Indicators			Static Sorption Capacity, mg/g			Static Sorption Capacity of Sulfate Titanium, mg/g (for Comparison)		
S_sp_, cm^2^/g	V_pore_, cm^3^/g	D, nm	Cs^+^	Sr^2+^	Co^2+^	Cs_+_	Sr^2+^	Co^2+^
81.0	0.18	8.5	148.9	79.2	69.3	195	272	66

## Data Availability

Not applicable.

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
