# Peer review of "Synthesis of Sorption Materials from Low Grade Titanium Raw Materials"

_materials, 2022, doi:10.3390/ma15051922_

Round 1

Reviewer 1 Report

The article “Thermochemical methods for the processing of low-titanium raw materials in acid media” reports the cleaning of sphenite ore to produce titanium-containing sorbents. The work lacks novelty since several results were already published in previous papers, some of them not cited. I suggest that the authors re-organize the paper to show the novelties of their work, remove the already published results, and cite the proper references. Among them, I found the following issues:

  1. Figure 5 is already present in two already published papers (doi:10.3390/min8100446 as Figure 6, doi:10.3390/ma13071599 as Figure 7), one of them not cited in the text.
  2. Figure 9 is already present in two already published papers (doi:10.3390/min9050315 as Figure 5, DOI: 10.1134/S0020168520110011 as Figure 1, doi:10.3390/ma13071599 as Figure 8), all of them not cited in the text.
  3. Figure 12 is already present in an already published paper (DOI: 10.1134/S0020168520110011 as Figure 3), not cited in the text.

Author Response

Thanks to the reviewer for carefully reading the article

The article as presented has never been published before. The authors tried to present the generalized material in its entirety.

The materials of the article contain specific data on the characterization and use of low-titanium raw materials containing the mineral titanite (sphene), which is currently a waste product from the processing of complex apatite-nepheline ore and goes to dumps after its enrichment.The authors propose two schemes for the processing of multicomponent mineral waste, allowing first to obtain titanium-containing precursors purified from impurity minerals, and then, on their basis, to synthesize, in particular, titanium-containing sorbents, which differ in composition and properties.Judging by the well-known literary sources, the obtained sorbents can be widely used in technologies for the purification of liquid effluents from toxic substances and radionuclides.The novelty of these technological schemes and the products obtained from them is confirmed by several patents cited by the authors in the list of publications.

Reviewer 2 Report

The paper is interesting and can be a good contribution for the Materials readers, but I think that it can be improved.

First of all, the paper structure can be improved. And in the title must be that this treatment is to obtain precursor for effective sorbents.

For example, in introduction the section 1.1 Characteristics of raw material sources must be in the Materials and Methods Section.

And also in Section materials and methods, it is important to have more sections, one for the characteristics of ram material, the other for the acid treatment, other for characterization of obtained materials, and other for  the synthesis of titanium adsorbent and then adsorption experiments or measurements.

In the results section, divide also in sections to organize the results. First acid treatments, and then titanium adsorbent synthesis and application.

The conclusion and references are ok.

Some Suggestions and queries:

1) What wastes are produced in acid treatment? How to discard them?

2) From Table 2 is not clear that HCl is better than H2SO4.  And the authors say that the efficiency of removing apatite and nepheline impurities from SO increases with the use of HCl. Explain that, please.

3) All XRD figures must be indexed with miller indexes or put the number of XRD charts.

4) It is interesting to see the N2 adsorption isotherms of materials. Put it at least on supplementary material.

5) For the sorbent synthesized, what is important? Superficial area? Composition? What moves the adsorption property?

6) In the introduction in lines 87 and 88, the authors say that Titanosilicate sorbents have superior ion exchange capacity than zeolites. Is it true for your synthesized sorbent? How does it work compared with zeolite material?

Author Response

Thanks to the reviewer for carefully reading the article

Reviewer 3 Report

This paper studied the conditions of acid treatment of unconventional titanium-containing raw materials to obtain precursors for the synthesis of effective sorbents. It is an interesting content, but arranged structure needs to be further improved. Therefore, it needs minor revision before it is published in this journal. The following issues should be carefully addressed.

  1. Authors can re-think about the title of the manuscript.
  2. Extensive English revisions for language and grammar are strongly needed.
  3. The research background is not well summarized, so the “Introduction” of this manuscript should be reorganized. Conventional titanium-containing raw materials have been pretreated by flotation before leaching, so it should be described in the “Introduction”, and several relevant references may be added to support this point, such as Trans. Nonferrous Met. Soc. China 31(2021) 3564−3578; Int. J. Min. Sci. Technol. 31 (2021) 1117–1128.
  4. Authors should provide experimental condition for their experiments.
  5. The necessity and innovation of the article should be presented at the end of introduction section.
  6. A lot of data about prediction analysis was disclosed, hence the significance, necessity and relevant works should be added.
  7. Much more explanations and interpretations must be added for the Results, which are not enough.
  8. The conclusions should be with key conclusions from the paper rather repeating experiments and results.

Author Response

(The authors gave the same response as above.)

Round 2

Reviewer 1 Report

The authors did not make any requested changes

Author Response

Thank you for your comments. The authors have added the characteristics of the sorption properties of the obtained sorbents and also added information on the raw materials for the sorbents. They also referred to previously printed materials.

Reviewer 2 Report

Thanks for the answers. I think that the paper is now more clear.  I didn't see the N2 isotherms curves and neither the supplemental material. I want to see that before my final recommendation.

Author Response

Thank you for your comments. Isotherms added to the text of the article

This manuscript is a resubmission of an earlier submission. The following is a list of the peer review reports and author responses from that submission.

Round 1

Reviewer 1 Report

This manuscript deals with processing of low-Ti raw materials for using the materials as sorbents. After leaching the raw materials with H2SO4 and HCl, respectively, the products were investigated with XRD and SEM, and were tested for sorbents. 

Unfortunately, this manuscript looks like project report, not academic article. There were many tests in this manuscript, but each test was not examined in details. e.g. only two leaching temperature and three pulp densities were investigated. The results for absorption test were not explained in details. I can't find what point was improved in this study. e.g. the absorption capacity should be compared with conventional or commercial materials. 

Therefore, I think this manuscript is not proper for publication. 

Reviewer 2 Report

Review comments

This paper proposed and investigated a series of processes for the apatite-nepheline ore processing waste, withα-TiP and αTi(HPO4)2∙H2O sorbents as the products. Overall, the contents in this manuscript are of great importance on the utilization of low-grade titanium ores and give a very comprehensive description of the experiments and analysis of the data acquired. I am convinced that the methods and data shown in this research are of high value to the understanding of the low-titanium mineral raw material management, at least for the ANO at Khibiny deposits specifically. However, some issues need to be addressed before the manuscript can be accepted.

  1. The manuscript is too long with so many details in it, which makes it hard to focus on the most important findings of this research. Maybe the authors can trim the article a little bit, such as putting the experimental procedures into the supporting information.
  2. One question, are the methods used in this research unique and original from the authors? The author should emphasize the innovative point of their research.
  3. Section 1.1. Characteristics of the raw material source should be inserted in Section 2 Materials and Methods.
  4. The value and use of the final sorbent materials should be introduced in the introduction.
  5. Some figures can be combined, such as Figure 11 and 12, Figure 13 and 14, and Figure 15 and 16, to put the relevant information together.
  6. Some other queries and suggestions:
    1. In Figure 3, what is the meaning of the unit Sm?
    2. Ln 201, 205…: The temperature unit C should be ℃.
    3. Ln 232: Should “BEI” be “BET”?
    4. Equation (5): please type the equation as other equations rather than using a picture.